# Allelopathic Effects of Sugarcane Leaves: Optimal Extraction Solvent, Partial Separation of Allelopathic Active Fractions, and Herbicidal Activities

**DOI:** 10.3390/plants13152085

**Published:** 2024-07-27

**Authors:** Ramida Krumsri, Hisashi Kato-Noguchi, Thanatsan Poonpaiboonpipat

**Affiliations:** 1Department of Agronomy, Faculty of Agriculture, Kasetsart University, Bangkok 10900, Thailand; ramida.kr@ku.ac.th; 2Department of Agricultural Science, Faculty of Natural Resources and Environment, Naresuan University, Phitsanulok 65000, Thailand; 3Department of Applied Biological Science, Faculty of Agriculture, Kagawa University, Miki 761-0795, Kagawa, Japan; kato.hisashi@kagawa-u.ac.jp; 4Center of Knowledge and Technology for Cane and Sugar, Faculty of Agro-Industry, Kasetsart University, 50 Ngamwongwan Road, Ladyao, Chatuchak, Bangkok 10900, Thailand

**Keywords:** allelopathy, sugarcane leaves, optimal extraction solvents, natural herbicides, physiological mechanism

## Abstract

The inhibitory potential of allelopathic plants is the subject of increasing research attention for their application in weed management. The sugarcane leaf is an agricultural waste product that has been reported to have allelopathic potential. Therefore, the present study determined the optimal organic solvent system and fractionation procedure to enhance the quantity of this extract and its allelopathic efficiency. Sugarcane leaves were extracted using five ethanol/water solvent ratios (*v*/*v*): 00:100, 25:75, 50:50, 75:25, and 100:00. Their allelopathic effects on seed germination and seedling growth were assayed in two major weeds, *Echinochloa crus-galli* (L.) Beauv. and *Amaranthus viridis* L. The results showed that the extract concentration, solvent ratio, and their interaction significantly inhibited the growth parameters in *A. viridis*. Consequently, a crude ethanol/water ratio of 00:100 was used to separate the active fraction via acid–base solvent partitioning. The acidic fraction (AE) exerted the greatest inhibitory effect and completely (100%) inhibited *A. viridis* at all concentrations, followed by the original crude fraction, neutral fraction, and aqueous fraction. Moreover, all of the fractions had selective effects, inhibiting *A. viridis* much more than *E. crus-galli* in the laboratory tests. The chemical analysis using gas chromatography/mass spectrometry indicated that the AE fraction contained 20 different compounds. The five major compounds included alkaloids, organic acids, and phenols. Therefore, the AE fraction was selected for formulation in a concentrated suspension and tested for its herbicidal characteristics. The formulation exhibited early post-emergence activities and had a stronger effect on *A. viridis* compared to *E. crus-galli*. The physiological mechanism of the formulation was tested against *A. viridis*. The thiobarbituric acid reactive substances and H_2_O_2_ occurred in the *A. viridis* leaf, which suggests lipid peroxidation and cell disruption.

## 1. Introduction

Weeds significantly reduce crop growth, yield, and quality, which results in economic losses. Therefore, weed management represents a significant element within crop management. Synthetic herbicides are a popular weed management tool because of their effectiveness, convenience, and affordability. However, the continuous use of herbicides has negative effects on human health and the environment and leads to plant resistance [1,2]. These concerns have resulted in an increase in eco-friendly crop production based on the use of natural products from plants or microorganisms [3]. The need for safer products with minimal effects on humans, animals, and agricultural output has been recognized, leading to less environmental pollution. For instance, allelochemicals enhance sustainable agriculture through the replacement of and reduction in the use of synthetic pesticides [4].

Many plant species, including crops and weeds, produce biologically active compounds or allelochemicals that inhibit plant growth and affect the physiological and biochemical processes of surrounding plants. Phenolic compounds and flavonoids are major allelochemicals found in plants [5]. The inhibitory effects of allelochemicals have been likened to the actions of synthetic herbicides [6]. Numerous allelopathic mechanisms have been reported for these compounds, including action via photosynthesis, respiration, disruption of plasma membrane integrity, water relationships, hormone interactions, and enzyme effects [7]. The crude extracts of plant materials are the primary products from extraction. These crude contents contain a significant amount of metabolites, depending on the source of the plant material and the extraction process [8]. The conceptual use of crude plant extracts as herbicides has been reported because crude extracts always have joint actions [9]. The crude extract of the bark of *A. altissima*, a tree in the family Simaroubaceae, contains ailanthone and has been assessed for pre- and post-emergence efficacy. That extract demonstrated complete control of *Eragrostis cilianensis* (Bellardi) Vignolo ex Janch and reduced the density of *Chenopodium album* L., *Mollugo verticillata* L., and *Solanum nigrum* L. [10]. Nevertheless, almost all crude plant extracts exhibit a limited ability to dissolve in water. Consequently, the extract necessitates converting it into a suitable blend procedure for application in practice [11]. For example, a crude aqueous ethanolic extract of *Tagetes erecta* L. was partially purified to eliminate inactive compounds and to formulate a soluble concentrated product that was contented surfactant and solvent. This product inhibits the emergence and growth of the weed *Echinochloa crus-galli* (L.) Beauv. when applied as a pre-emergent herbicide [12]. The crude extracts of *Chromolaena odorata* (L.) King and Robinson, isolated using hexene, have been formulated into a concentrated suspension. This formulation demonstrates greater efficacy in weed control at the early-post emergence stage when used at the pre-emergence stage [13]. 

Nevertheless, the cost, yield, and allelopathic activity of the formulated products from crude extracts must be considered. Organic solvent extraction methods are the most common, as they produce a good yield of lipid compounds [14]. The extract yields and allelopathic activities of the plant material are strongly dependent on the nature of the extracting solvent due to the presence of different chemical characteristics and polarities that may or may not be soluble in a particular solvent. Polar solvents are frequently used to recover polyphenols from a plant matrix. The most suitable of these solvents are aqueous mixtures containing ethanol, methanol, acetone, and ethyl acetate [15]. Methanol and ethanol have been extensively used to extract allelopathic compounds from various plants. Some studies have demonstrated the high efficacy of aqueous methanol for recovering phenolic compounds from rice bran [16] and *Moringa oleifera* Lam. leaves [17]. In contrast, the leaf extracts of *Thunbergia laurifolia* Linn. obtained with a 0:100 (water/ethanol) solvent had the greatest inhibitory effect on *Amaranthus gracilis* L., and the yields of the crude extracts decreased in the order of 75:25 > 50:50 > 75:25 > 100:00 > 0:100 (water/ethanol) [18]. Therefore, selecting the appropriate solvent system is important for obtaining a high allelopathic potential and a high yield from a crude plant extract.

Sugarcane is an important cash crop in the family Gramineae, which is cultivated in tropical and sub-tropical regions worldwide [19]. Sugarcane is used to produce sugar, which contributes about 92% of the sugar produced [20]. Brazil, India, Australia, China, and Thailand produce about 40% of the world’s sugar. Of the 115 sugar-producing countries, 67 countries cultivate sugarcane alone, 39 grow sugar beets alone, and 9 grow both; thus, sugarcane contributes about 70% of global sugar production [21]. In developing countries, a large amount of sugarcane field residue is traditionally burned, which has negative effects on air quality and emits a range of atmospheric pollutants (e.g., PM_10_, PM_2.5_, NOx, and SO_2_) [22,23]. Therefore, the challenge is to identify a use for sugarcane leaf residue. Many studies have indicated that sugarcane field residue has an allelopathic activity on plant growth. For instance, Viator [24] reported that sugarcane field residue leachates from two soil types reduced the germination and radical growth of (*Avena nuda* L.) and rye (*Secale cereale* L.) but not tomato (*Solanum lycopersicum* L.). Sugarcane straw phytotoxins are a major concern because the straw is commonly left on the soil surface after harvest in Argentina and other countries [25]. Sugarcane straw leachates inhibited weed growth in field experiments [26]. Moreover, an aqueous extract of sugarcane leaf cultivar 51 demonstrated allelopathic effects on the germination and seedling biomass of wheat (*Triticum aestivum* L.) [27]. 2,4-Dihydroxy-1,4-benzoxazin-3-one, 2-benzoxazolinone, trans-ferulic acid, cis-ferulic acid, vanillic acid, and syringic acid were isolated from sugarcane leaves [25,28]. As a large quantity of sugarcane leaf residue could be used due to its allelopathic effects, sugarcane leaves are a good source for developing bioherbicides. Extracting allelopathic compounds from sugarcane leaf extracts requires appropriate extraction methods and techniques to provide bioactive ingredient-rich crude extracts for developing bioherbicides. 

Therefore, the present study was conducted to (i) identify an organic solvent extraction system for sugarcane leaves, (ii) separate the active fraction via acid–base solvent partitioning to enhance the quantity of the extracts and their allelopathic efficiency, (iii) identify the active compounds from fractions separated with GC-MS analysis to better understand their allelopathic effects, (iv) evaluate the herbicidal characteristic as early post-emergence of the selected crude fraction under glasshouse conditions, and (v) explore the metabolic processes involved in cellular leakage, lipid peroxidation, as well as hydrogen peroxide against *Amaranthus viridis* L.

## 2. Results

### 2.1. Allelopathic Effects of Solvent Extracts against Weeds and the Crude Extract Yields 

The different extraction solvent systems significantly affected the germination and growth of *Amaranthus viridis* (Figure 1). The degree of inhibition differed among the crude extracts and the tested concentrations. The ethanol extracts exerted highly significant allelopathic inhibitory effects on all *A. viridis* growth parameters; the extracts using the water/ethanol ratios of 50:50 and 25:75 exerted intermediate inhibitory effects and the extracts using the water/ethanol ratios of 100:00 and 75:25 caused the least severe effects. At an extract concentration of 10,000 ppm, the growth of *A. viridis* seedlings decreased when treated with the crude extracts obtained from the 100:00, 50:50, and 25:75 water/ethanol solvents, while the other ratios decreased *A. viridis* seedling growth by more than 80% compared to the control (distilled water). Based on these results, the order of inhibition of *A. viridis*, using the extracts with the indicated water/ethanol ratios, was 00:100 (1565–3160 ppm) > 25:75 (2990–3434 ppm) > 50:50 (3094–4819 ppm) > 75:50 (3864–3850 ppm) > 100:00 (4997–6298 ppm) (Table 1). Moreover, the IC_50_ (50% inhibitory concentration) values of all extracts indicated that the degree of inhibition of the growth parameters followed the order of root length > shoot length > seed germination. In contrast, no significant inhibitory effect on seed germination or seedling growth was observed for *Echinochloa crus-galli* (Figure 1). Moreover, the yield of the crude extract depended on the proportion of solvent used; the yield increased as the ethanol concentration increased (Table 1). The water/ethanol extraction ratios of 25:75 and 00:100 produced the highest crude extract yields, while 100:00 produced the lowest yield.

### 2.2. Allelopathic Effects of the Fractions Extracted by Solvent Partitioning

To increase the activity in the crude water/ethanol (100:00 [*v*/*v*]) (OR) fraction, the bioactive fraction was separated via acid–base solvent partitioning into the aqueous (AQ), neutral (NE), and acidic (AE) fractions. The allelopathic effects of all separated fractions and the OR fraction were determined on *A. viridis* and *E. crus-galli* seed germination and seedling growth. After partitioning, the inhibitory activity in the separated fractions increased compared to the OR fraction (Figure 2 and Figure 3). The degree of inhibition in each fraction increased with increasing extract concentration. At a concentration of 1250 ppm, the AE fraction had maximal activity and completely inhibited *A. viridis* germination and seedling growth; meanwhile, the other fractions had no effects compared to the control. The highest concentrations of the AE and OR fractions completely inhibited *A. viridis* seedling growth, whereas the NE and AQ fractions inhibited more than 70% compared to the control.

### 2.3. GC-MS Analysis of the Acidic Fractions Separated from Sugarcane Leaf Extracts

The bioactive components in the acidic fractions separated from the sugarcane leaf extracts were identified by GC-MS (Figure 4). A total of 20 compounds were identified with varying peaks, retention times, and peak areas (in terms of compositional percentages) (Table 2). 

The following compounds were detected which arranged in a decreasing order based on their mass: iron, tricarbonyl l[N-(phenyl-2-pyridinylmethylene) benzenamine-N, N-] (24.50%), dibenzylamine (22.96%), 1,2,4,3,5-trioxadiborolane, 3,5-dimethyl (16.19%), phenol, 2,4-bis (1,1-dimethylethyl) (10.48%), benzothiazole, 2-(2-hydroxyethylthio) (10.09%), cyclohexyl methyl S-2-(dimethylamino) ethyl propyl phosphonothiolate (3.08%), 3-(methylthio) phenyl isothiocyanate (2.48%), bis (2-ethylhexyl) hydrogen phosphite (2.23%), propane, 1-[1 [difluoro[(trifluoroethenyl)oxy]methyl]-1,2,2,2-tetrafluoroethoxy]-1,1,2,2,3,3,3-heptafluoro (1.01%), 3-methyl pyrazolobis (diethylboryl) hydroxide (0.78%), 2H-pyran-2-one, 6-[4,4-bis(methylthio)-1,2,3-butatrienyl] (0.75%), 1H-1,2,3,4-tetrazole, 1-(4-methoxyphenyl)-5-[(phenylmethyl)sulfonyl] (0.74%), cyclohexyl methyl silane (0.64%), tris (tert-butyldimethylsilyloxy) arsane (0.63%), arsenous acid, tris (trimethylsilyl) ester (0.62%), pyridine, 2-chloro-3-fluoro-, 1-oxide (0.57%), borinic acid, diethyl-, (2-ethyl-1,3,2-dioxaborolan-4-yl)methyl ester (0.48%), O, O-dimethyl [1-(4-methyl-1,2,5-oxadiazol-3-ylamino)-1-(2-fluorophenyl) methyl] phosphonate (0.46%), 2,6-bis (thiocyanatomethyl)-4-methylanisole (0.33%), and caprolactone oxime, (NB)-O-[(diethylboryloxy) (ethyl) boryl] (0.32%).

### 2.4. Investigation of AE Formulation Herbicidal Activity in Pot Experiments

The AE formulation was spray-applied to test for efficacy as an early post-emergence herbicide on seedlings of *A. viridis* and *E. crus-galli*. After 24 h of spraying, the *A. viridis* seedlings exhibited indications of mortality, including collapse and wilting (Figure 5B). Nevertheless, the formulation exhibited minimal impact on *E. crus-galli* when applied at equivalent concentrations (Figure 5A). The concentrations of 50 and 25 mg/mL led to complete mortality of *A*. *viridis* in 100% and 80% of cases, respectively (Figure 6B). In contrast, the concentration of 12.5 mg/mL resulted in a death rate of 20%, as shown in Figure 5B. Among the *E. crus-galli* seedlings, only the treatment with a concentration of 50 mg/mL caused a 20% effect, while the other treatments had no impact (Figure 6A). The dry weight of the weeds was found to be closely correlated and comparable to the death rate (Figure 6A,B). 

### 2.5. AE Formulation Induced Oxidative Damage

*Amaranthus viridis* leaves treated with the AE formulation showed increased malondialdehyde (MDA) content (Figure 7A), thereby indicating enhanced lipid peroxidation. The concentration of MDA showed a significant increase of 71.4% and 83.3% when compared to water, at concentrations of 25 and 50 mg/mL, respectively (Figure 7A). There was no difference in the amount of MDA content at 12.5 mg/mL. This corresponds with the findings of the hydrogen peroxide (H_2_O_2_) levels presented in Figure 7B. A significant rise was seen in the leaves that were sprayed with 25 and 50 mg/mL, but no significant change was observed at 12.5 mg/mL. The concentration of H_2_O_2_ exhibited a proportional increase at higher values. In addition, the results further showed that the AE formulation caused an electrolyte leakage (EL) in the treated leaves, and there was a significant increase in EL at 25 and 50 mg/mL (Figure 7C).

## 3. Discussion

The sugarcane leaf extracts obtained from the 00:100 water/ethanol ratio inhibited *Amaranthus viridis* seed germination and seedling growth more than the water/ethanol ratios of 25:75, 50:50, and 75:25. This finding is in line with the findings in the studies by Xavier and Arun [29] and Mugao [30], who reported higher efficacies of ethanol as a solvent because of its polarity, resulting in a higher concentration of secondary metabolites [31]. The low efficacy of aqueous plant extracts is due to the inability of water to extract nonpolar active compounds from the plant material [32]. Additionally, different solvent systems used to extract the plant material result in different extraction efficacies depending on the composition of each particular plant [33]. Our results were similar to those of Thinh [16], who noted that the *T. laurifolia* leaf extract obtained with 0:100 water/ethanol as the solvent showed the greatest inhibitory effect on *A. gracilis*; contrary to the effects of the crude extract, they found that the crude extract yields of various solvents decreased in the order of 75:25 > 50:50 > 25:75 > 100:00 > 0:100 (water/ethanol). However, Li [34] reported that the water-soluble extract of *Artemisia argyi* Lévl. et Van. exhibited the strongest inhibitory effect on the seed germination and seedling growth of *Brassica pekinensis* (Lour.) Rupr. and *Lactuca sativa* L., compared to extracts obtained using 50:50 and 0:100 water/ethanol ratios. Six main compounds were identified in the water extracts of *A. argyi*, including caffeic acid, schaftoside, 4-caffeoylquinic acid, 5-caffeoylquinic acid, 3,5-dicaffeoylquinic acid, and 3-caffeoylquinic acid. The main compounds extracted with the 50:50 water/ethanol solvent were 4,5-dicaffeoylquinic acid, schaftoside, rutin, kaempferol 3-rutinoside, and eupatilin. Such inhibitory effects may be why different extracts of the same plant material often contain diverse components, resulting in different effects [35,36]. Thus, the extracts prepared from different solvents had varying degrees of inhibitory activity. These results indicate that selective extraction from natural sources using an appropriate solvent system is important for obtaining fractions with a high allelopathic potential and a high yield. Plant extracts comprise a mixture of various bioactive and inactive compounds, with differences in the solubility of the extractives [37,38].

The crude extract of the 0:100 water/ethanol ratios was also employed in the primary isolation of the active compound using an acid–base solvent partitioning technique. These results indicated that the majority of the allelopathic compounds produced by the sugarcane leaf extracts contained an acidic fraction. This result is similar to those of Teerarak [39], who reported that the AE fraction separating *Jasminum officinale* var. *grandiforum* was more inhibitory than the other fractions, and the allelopathic compound was identified as a secoiridoid glucoside called oleuropein. Furthermore, the selective effects of all separated OR fractions from sugarcane leaves were observed on the growth of different plant species. At the highest concentrations, all fractions suppressed the seed germination and shoot and root growth of *A. viridis*.

In contrast, the AE and NE fractions only slightly inhibited *Echinochloa crus-galli* root growth, and the remaining fractions had no effects on *E. crus-galli* at any of the concentrations tested (Figure 2). These results agree with several studies. The response of plant species varies, depending on the type of allelopathic substances in a given plant extract [40,41]. Based on our results, the AE fraction separated from the sugarcane leaf extract was a selective inhibitor, especially in a monocotyledonous plant species. Although further assessments are needed, this semi-selective assay may be useful for the development of herbicides to control weed growth.

Among the identified compounds, the five main compounds detected in the AE fraction were alkaloids, organic acids, or phenols. Compound (**1**) iron, tricarbonyl l[N-(phenyl-2-pyridinylmethylene) benzenamine-N, N-] (24.5%), was an alkaloid and compound (**2**), 3,5-dimethyl-1,2,4,3,5-trioxadiborolane (16.19%), was an organic acid. The other three compounds, (**3**) phenol, 2, 4-bis (1,1-dimethylethyl) (10.48%), (**4**) dibenzylamine (22.96%), and (**5**) benzothiazole, 2-(2-hydroxyethylthio) (10.09%), were phenolic compounds. These compounds have been previously detected in different plant species and reported as playing roles in many biological activities [42,43,44]. Commonly, the compound cyclohexyl methyl silane, borinic acid (**6**), diethyl-, (2-ethyl-1,3,2-dioxaborolan-4-yl) methyl ester (**9**), arsenous acid, tris (trimethylsilyl) ester (**18**), and tris (tert-butyldimethylsilyloxy) arsane (**20**) are organosilicon and organoboron compounds that contain elements such as silicon, boron, and arsenic [45,46,47,48]. They are applied in organic synthesis, catalysis, materials science, and pesticides. Our experiment revealed the presence of these compounds in sugarcane leaf extracts. Consequently, the samples may retain residual pesticides even after harvest. We collected sugarcane leaves from an agricultural field for experiments; in practice, farmers use pesticides.

In the present study, the chemical profile of the extracts disclosed a high proportion of phenolics (>43%) (Table 1). Phenolics are an important class of allelopathic chemicals with a wide range of actions [49,50]. Jia [51] discovered a synergistic allelopathic activity in mixtures of phenolic compounds, which were dependent on the interactions between the concentration, combination of compounds, and test species sensitivity. Bashar [52] reported that a *Parthenium hysterophorus* L. extract had a higher inhibitory effect than its individual compounds. The more potent inhibitory effects of the extract could be due to unique chemical combinations that work in an additive or synergistic manner [53]. Therefore, we speculate that these chemical combinations are responsible for the overall allelopathic effects of the extracts. 

The AE fraction was subsequently transformed into a suspension concentrate. During the initial investigation, the formulation exhibited stronger early post-emergence activity compared to its pre- and post-emergence characteristics. In this report, we discussed the impact of early post-emergence. The AE formulation had a more significant phytotoxic effect on the *A. viridis* seedlings compared to *E. crus-galli* at the same dose. Therefore, *A. viridis* was chosen as the model weed for studying the physiological mechanisms. The thiobarbituric acid reactive substance (TBARS) assay is the most often utilized test for measuring the concentration of malondialdehyde (MDA), which is the end-product of lipid peroxidation [54]. The findings indicated that the sample leaves of *A. viridis* treated with the AE formulation exhibited an increase in TBARS content at higher concentrations, suggesting the occurrence of lipid peroxidation (Figure 6). This observation that the formulation of crude extraction spraying on the *E. crus-galli* leaves exhibited the accumulation of TBARS was similar to previous reports regarding the crude hexane fraction from *Chromolaena odorata* (L.) R.M.King and H.Rob leaves [13]. In addition, we analyzed the treated leaves, which revealed an increase in the concentration of hydrogen peroxide. This suggests an accumulation of reactive oxygen species, which, in turn, leads to heightened oxidative stress and disruption of cellular metabolic functions. H_2_O_2_ is a potent oxidizing agent that can cause localized oxidative damage, resulting in the disruption of metabolic activity and the loss of cellular integrity in the areas where it builds up [55,56]. The observed rise in electrolyte leakage (EL) in sprayed *A. viridis* leaves suggests membrane damage because EL is an accurate indicator of membrane damage. The damage occurs as a result of the peroxidation of cell membranes caused by an oxidative burst [57] that is related to the enhancement in the MDA and H_2_O_2_ content. 

## 4. Materials and Methods

Site description

The experiments were conducted in the weed science laboratory of the Department of Agricultural Sciences, Faculty of Agriculture, University of Naresuan, Thailand.

Plant material

Sugarcane leaves (variety Khonkaen 3) were collected from an agricultural field in Phitsanulok Province, Thailand, in December 2021. Fresh sugarcane leaves lacking pathogenic and insect damage were harvested at the mature stage (9 months after sowing) and irrigated with tap water to remove dust. The materials were dried in a hot-air oven at 45 °C for 72 h and ground into a powder using an electric grinder.

Test plant species

*Echinochloa crus-galli* and *Amaranthus viridis* seeds were collected from agricultural fields in Phitsanulok Province, Thailand. The *E. crus-galli* seeds were aired under full light for 72 h and incubated in a hot-air oven at 50 °C for 24 h to break their dormancy for future experiments. 

### 4.1. Allelopathic Effects of Solvent Extracts against Weeds and the Crude Extract Yields

Sugarcane leaf powder was extracted to determine their allelopathic effects following the protocol described previously by Thinh et al. [18]. A 20 g portion of leaf powder was extracted with 500 mL of distilled water and ethanol (100:0, 25:75, 50:50, 75:25, and 0:100 [*v*/*v*]) and left in the dark at room temperature (25 ± 2 °C) for 72 h. Each aliquot was extracted and filtered through a single layer of filter paper (No. 1; Whatman Inc., Clifton, NJ, USA). The residues were re-extracted under the same conditions as the initial extraction procedure and then filtered. The two supernatants were combined, and the solvent was removed in a rotary vacuum evaporator (BUCHI Rotavapor R255; BUCHI, Lausanne, Switzerland) at 40 °C to yield the crude aqueous ethanol extracts. The crude extracts were weighed and kept shaded from light and at a low temperature (4 °C). 

The crude sugarcane leaf extracts were dissolved in and diluted with their initial solvents to four final concentrations: 1250, 2500, 5000, and 10,000 ppm. Then, 5 mL of each concentration was added to filter paper (No. 2) in a 90 mm Petri dish. The treated filter papers were evaporated to dryness in a fume hood at room temperature (25 ± 2 °C). All treated filter papers were moistened with 5 mL of 500 ppm (*v*/*v*) aqueous solution of Tween 20. Twenty seeds of *A. viridis* and/or *E. crus-galli* were evenly spread onto the paper. An aqueous solution of Tween 20 without the extract was used as the control treatment. The bioassays were undertaken with four replicates. All Petri dishes were placed in a growth chamber at 25–32 °C with a 12 h/12 h dark/light photoperiod and light intensity (cool white, 840) of 100 μmol m^−2^ s^−1^ with a relative humidity of about 80%. After 7 days of incubation, the number of seeds that germinated and the shoot and root lengths were measured. The growth length data were calculated as a percentage of seedling length using the following formula: Percentage of seedling length (%) = (length of treatment/length of control) × 100

### 4.2. Solvent Partitioning of the Active Fractions and Bioassay

An acid–base solvent partitioning technique (Figure 8) was used to separate the active components in the sugarcane leaf extracts [12]. The sugarcane leaf extracts were prepared with the most effective solvent, as determined in Section 2.4. The extracts were evaporated to obtain the crude extract residue (original crude fraction; OR fraction). The crude residue was dissolved in 300 mL of distilled water and stirred vigorously at 45 °C for 20 min. The aqueous residue solution was acidified to a pH of 3 using 6 N HCl and against ethyl acetate (EtOAc) at a ratio of 1:1 (*v*/*v*), and this procedure was repeated three times; it was then divided into aqueous and EtOAc solutions. The aqueous solution was adjusted to a pH of 7 with NH_4_OH to produce the aqueous (AQ) fraction. The EtOAc solution was dried overnight with anhydrous MgSO_4_ and filtered through filter paper. The filtrate was concentrated to 200 mL and then repeated three times against an equal volume of saturated aqueous NaHCO_3_. The EtOAc solution was dried with MgSO_4_ and filtered to yield the EtOAc-soluble neutral (NE) fraction. The combined NaHCO_3_ solution was evaporated to 200 mL and adjusted to a pH of 3 with 6 N HCl against an equal volume of EtOAc, repeated 3 times. The EtOAc solution was dried with MgSO_4_ and filtered to yield the EtOAc-soluble acidic (AE) fraction. All separated fractions were concentrated via a reduction in the pressure to provide the yield of the OR, AQ, NE, and AE fractions. 

The aliquots of the separate fractions were redissolved and diluted with the original extraction solvent to compare their phytotoxic effects. Seed germination and seedling growth bioassays of the test plants and the calculated data followed the protocols previously described.

### 4.3. GC-MS Separation of Compounds in the Active Fractions of Sugarcane Leaf Extracts

The most active fraction was dissolved in and diluted with dichloromethane (HPLC grade) to obtain a concentration of 100 ppm. The sample solution was filtered using a syringe filter and held at 4 °C before analysis. The GC-MS analysis was performed using a model 6890 gas chromatograph (Agilent Technologies, Palo Alto, CA, USA) equipped with a mass-selective detector. A fused silica capillary HP-5 column (30 m × 0.25 mm i.d., 0.25 μm film thickness) was used for the separation. The procedure included a 40 °C initial temperature (2 min of hold time) and a 280 °C final temperature (20 min of a hold time) with a heating rate of 12 °C min^−1^, using the split mode (10:1 ratio) with a flow rate 3 mL min^−1^ and a 42 min total run time. The MassHunter software ver. 10.0.368 (Agilent) for GC-MS systems was used for molecular identification with the NIST 17 Mass Spectral Library. The relative contents of each component in the sample were evaluated through the normalization of the peak areas as a percentage of the total detected components.

### 4.4. Herbicidal Activities of the Active Fractions 

The AE fraction was chosen to produce the available product due to its superior activity. The aqueous suspension concentrate (SC) formulations were prepared by combining crude AE (active ingredient) at a concentration of 40%, NP-40 nonylphenol ethoxylate at a concentration of 5%, sodium lauryl sulfate at a concentration of 5%, talcum clay at a concentration of 20%, and deionized water to achieve a total concentration of 100%. This resulted in the AE crude extract containing 40% *w*/*v* of the active ingredient (ai). Prior to usage, the product was stored at an ambient temperature of 25–30 °C and shielded from light.

The herbicidal properties of an early post-emergence application of a spray containing the AE product (SC formulation) of sugarcane leaf extract were assessed. A plastic container with a radius of 10 cm and a height of 10 cm was filled with a mixture of peach moss and perlite in a ratio of 3 parts peach moss to 1 part perlite by weight. Ten germinating seeds of either *E. crus-galli* or *A. viridis* were planted at a depth of 2 mm. The *E. crus-galli* and *A. viridis* seedlings that emerged 7 days after planting at the two leaf stages were categorized as early post-emergence. The weeds were treated with various concentrations of active ingredients (ai), including 12.5, 25, and 50 mg/mL. The application was performed using a hand nozzle and a water spray at a rate of 750 L/ha. In the control group, distilled water was utilized for application. The pots were positioned within a greenhouse, where the temperatures ranged from 26 to 38 °C. The experiment was replicated four times, and the pots were planted in a completely randomized design. The symptoms of weed damage in the field were observed and documented at 24 h intervals for the period of one week following the application of the spray. Afterward, the shoot dry weight was collected from the surviving seedlings, which were dried in an oven at 80 °C. 

### 4.5. Estimating Membrane Injury

In order to examine the physiological impacts of the AE formulation on weeds, *A. viridis* was chosen as a representative weed species. We applied an AE formulation to the leaves of *A. viridis* plants 14 days after germination. The leaf samples were then collected 24 h after the application.

The membrane injury index was assessed using the method developed by Mehrabi [58], which involved measuring the electrical conductivity of leaf leachates in deionized water. In order to measure the electrolyte leakage (EL), 0.5 g of leaves were chopped and placed in a Falcon tube with 15 mL of distilled water. The tube was then placed on a shaker at a temperature of 25 °C for a duration of 24 h. Subsequently, the electrical conductivity of the samples (EC1) was measured using a conductivity meter. Next, the tubes were immersed in a water bath that was heated to 95 °C and kept there for 1 h. During this time, the electrical conductivity (EC2) was measured once more. 

Ultimately, the EL was determined using the following formula: EL (%) = EC1/EC2 × 100 
where EL = electrolyte leakage; EC1 = initial electrical conductivity; and EC2 = final electrical conductivity.

### 4.6. Lipid Peroxidation

Lipid peroxidation, which refers to the oxidative damage to lipids, was assessed by measuring the levels of malondialdehyde (MDA). The concentration of MDA, a significant marker of lipid peroxidation, was measured. The method was adapted from the works of Heath and Packer [59], Singh [60], Bahmani [61], and Taratima [62].

A total of 0.1 g of leaf tissue was mixed with 5.0 milliliters of trichloro ethanoic acid (TCA) solution, which had a concentration of 0.1% (*w*/*v*). The extracts were transferred into tubes and then subjected to centrifugation at a speed of 10,000 rpm for 10 min at a temperature of 4 °C. Then, 1 milliliter of the liquid above the extracted substances in the centrifuged tubes was combined with 4.0 milliliters of a solution containing 0.5% thiobarbituric acid (TBA) in 20% trichloroacetic acid (TCA). The solution was subjected to incubation at a temperature of 95 °C for a duration of 30 min. The reaction was rapidly halted by subjecting it to chilling on ice for a duration of 10 min. The tubes were subjected to centrifugation at a speed of 10,000 revolutions per minute for a duration of 10 min at a temperature of 4 °C. A total of 1.5 mL of the liquid portion (supernatant) above the extracted material in the centrifuged tubes was transferred into cuvette tubes. The supernatant’s absorption was quantified using a UV spectrophotometer at a wavelength of 532 nm. The measurement was adjusted to account for any non-specific absorption at 600 nm, compared to a blank containing all ingredients except for the plant sample.

The concentration of MDA was determined through the application of the Lambert–Beer law, utilizing an extinction coefficient (ε) of 155 mM/cm. The concentration was then given as nmol MDA per gram of fresh weight (FW).
Concentration of MDA (µmole/g FW) = ((A532 − A600) × Vf × Ve)/(155 × Va × FW)

A532 = the absorbance at 532 nm, A600 = the absorbance at 600 nm;Vf = final volume;Ve = volume of TCA used for extraction;Va = volume of supernatants used in absorbance detection;FW = fresh weight of samples.

### 4.7. Hydrogen Peroxide (H_2_O_2_) Content

The following method was derived from the research of Singh [60] and Velikova [63]. The leaf tissue (0.1 g) was crushed using a homogenizer with 5.0 mL of trichloroacetic acid (0.1%, *W*/*V*) in a container filled with ice. The resulting mixture was then transferred to tubes and subjected to centrifugation at a speed of 12,000 rpm for 15 min. Then, 0.5 mL of the supernatant was combined with 0.5 mL of phosphate buffer (pH 7.0) and 1.0 mL of potassium iodide (1 M). The mixture was then incubated at room temperature in a dark place for 1 h. Then, the 1.5 mL of the liquid remaining after sedimentation was added to the cuvette tubes. The supernatant’s absorbance was quantified using a UV spectrophotometer at a wavelength of 390 nm, compared to all the constituents, excluding the plant sample as a reference. The H_2_O_2_ concentration was measured using an extinction coefficient (ε) of 0.28 mM/cm and reported as µmol/g FW.
Concentration of H_2_O_2_ (µmole/g FW) = (A390 × Vf × Ve)/(0.28 × Va × FW) 

A390 = the absorbance at 390 nm;Vf = final volume;Ve = volume of TCA used for extraction;Va = volume of supernatants used in absorbance detection;FW = fresh weight of samples.


*Statistical Analysis*


All bioassays were conducted using a completely randomized design with four replicates. The data from all experiments were analyzed using Tukey’s test in SPSS software (ver. 21.0; IBM Corp., Armonk, NY, USA). GraphPad Prism ver. 6.0 (GraphPad Software Inc., La Jolla, CA, USA) was used to identify the 50% inhibitory concentrations of the tested fractions.

## 5. Conclusions

The results of this study revealed that the sugarcane leaf extracts derived using different solvents resulted in different allelopathic effects. The ethanol extract had the most deleterious effects on the germination and growth of *A. viridis* compared to the other three extraction solvents used in this study. The water/ethanol solvent extraction ratios of 25:75 and 00:100 produced the highest crude extract yields, while the 100:00 water/ethanol ratio produced the lowest yield. Therefore, ethanol should be used as an extraction solvent if an inhibitory effect is desired. The active fraction in the ethanol crude extract (OR) was separated via acid–base solvent partitioning to enhance the allelopathic efficiency. After partitioning, the inhibitory activity of the separated fraction increased compared to the OR fraction. The AE fraction exhibited a maximal inhibitory activity on *A. viridis* compared to the other fractions. All fractions exhibited greater inhibitory effects in *A. viridis* than in *E. crus-galli*. The five major compounds in the AE fraction, as detected by GC-MS, were iron, tricarbonyl l[N-(phenyl-2-pyridinylmethylene) benzenamine-N, N-]; 3,5-dimethyl-1,2,4,3,5-trioxadiborolane; phenol, 2, 4-bis (1,1-dimethylethyl); dibenzylamine; and benzothiazole, 2-(2-hydroxyethylthio). Based on our findings, the optimized solvent extraction with ethanol increased the inhibitory activity of the sugarcane leaf extract. 

Furthermore, the acidic fraction (AE) obtained from the crude ethanol extract was chosen for formulation in a concentrated suspension due to its potent inhibitory and specific effects, and the formulation was evaluated for its herbicidal properties. The formulation exhibited higher efficacy in the early stages of plant growth after emergence rather than before emergence. It also demonstrated a greater effect on *A. viridis* than *E. crus-galli*. The formulation resulted in the death of weeds through the process of collapse and wilting. The physiological mechanism of the formulation was evaluated in relation to its effects on *A. viridis*. The presence of thiobarbituric acid reactive chemicals and H_2_O_2_ in the *A. viridis* leaf indicated the occurrence of lipid peroxidation and cell disruption, leading to electrolyte leakage and cell death.

## Figures and Tables

**Figure 1 plants-13-02085-f001:**
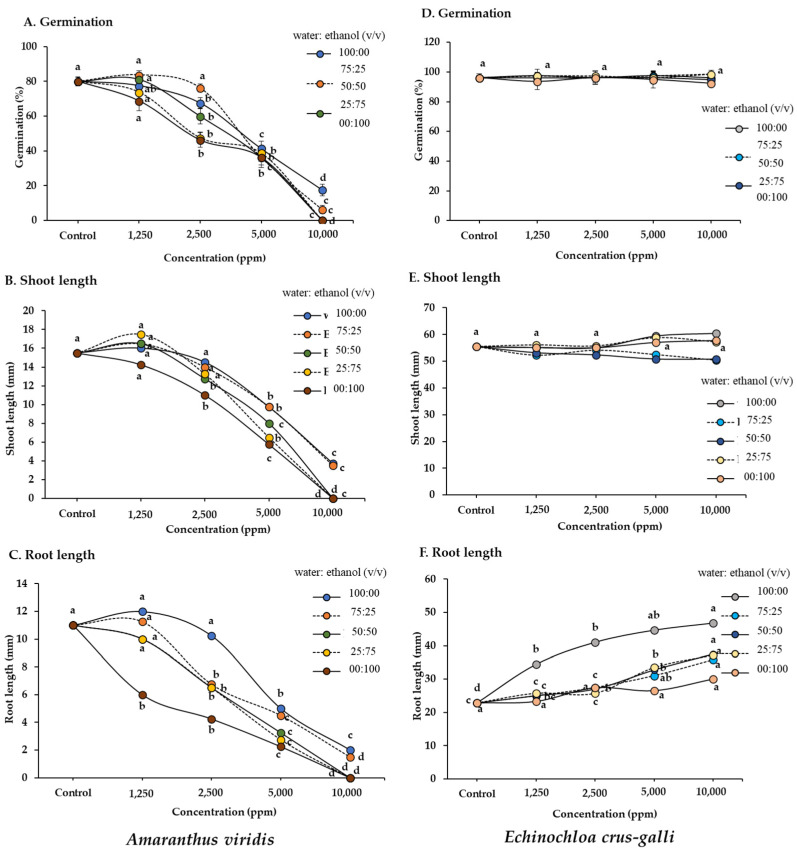
Effects of crude extracts of sugarcane leaves obtained from various solvents at different concentrations on the germination and growth of (**A**–**C**) *Amaranthus viridis* and (**D**–**F**) *Echinochloa crus-galli*. The results are presented as the means of four replicates. Different letters above each point indicate a significant difference (Tukey’s HSD, *p* < 0.05).

**Figure 2 plants-13-02085-f002:**
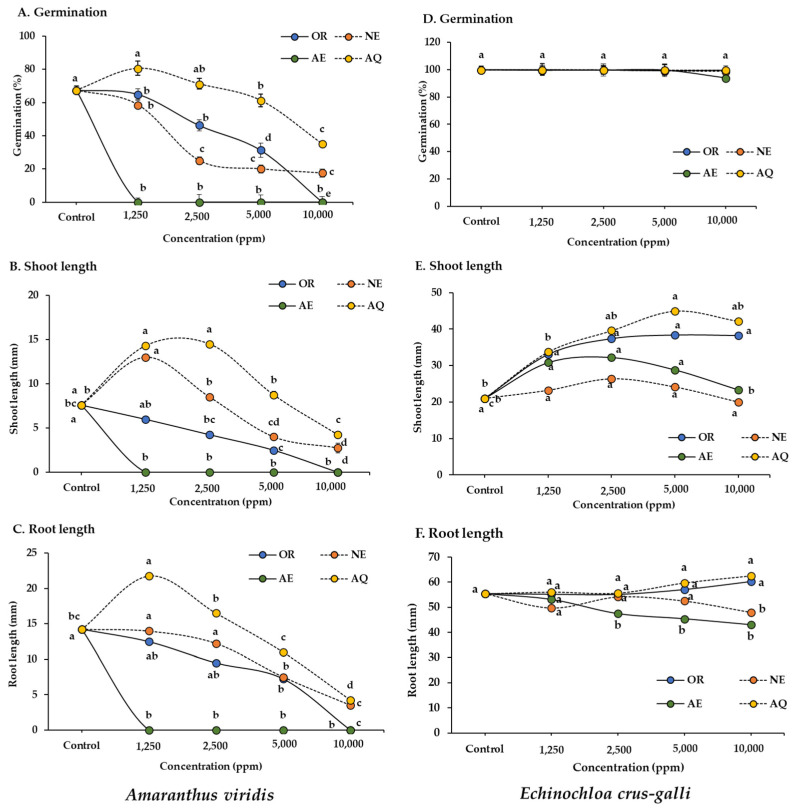
Effects of the ethanol crude extract (OR) and separated fractions—aqueous (AQ), neutral (NE), and acidic (AE) fractions—at different concentrations on seed germination and seedling growth in *Amaranthus viridis* and *Echinochloa crus-galli*. The results are the means of four replicates. Different letters above each point indicate a significant difference (Tukey’s HSD, *p* < 0.05).

**Figure 3 plants-13-02085-f003:**
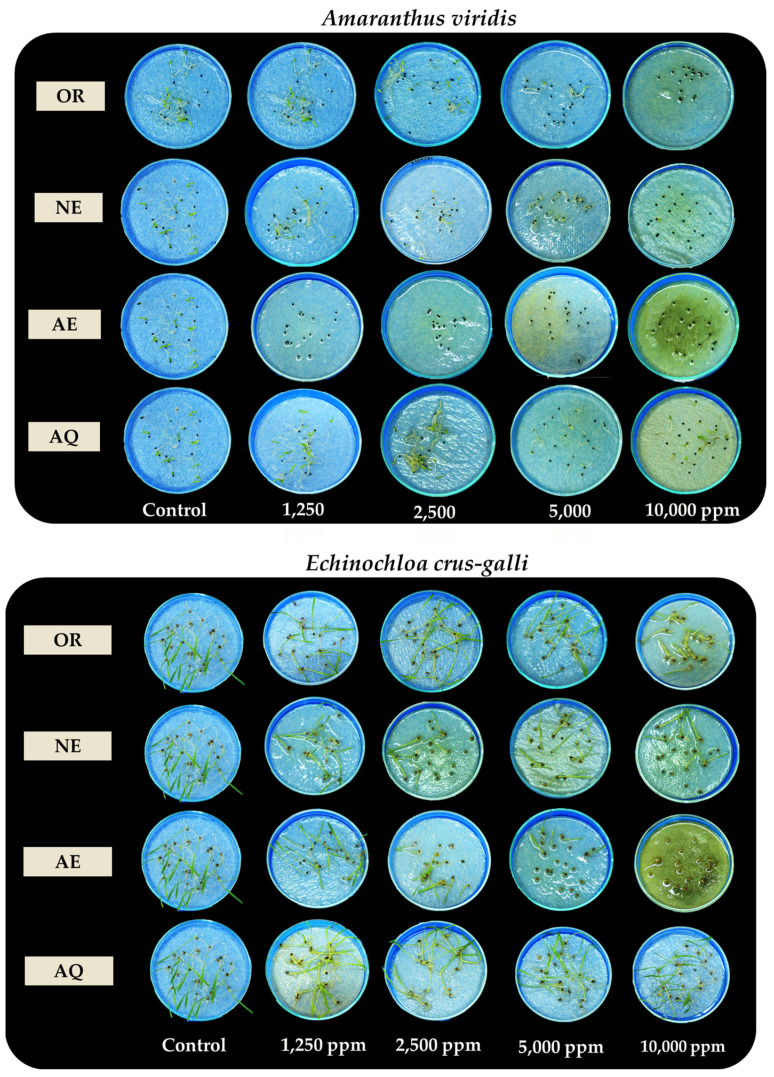
Effects of the ethanol crude extract (OR) and separated fractions—aqueous (AQ), neutral (NE), and acidic (AE) fractions—at different concentrations on seed germination and seedling growth in *Amaranthus viridis* and *Echinochloa crus-galli*.

**Figure 4 plants-13-02085-f004:**
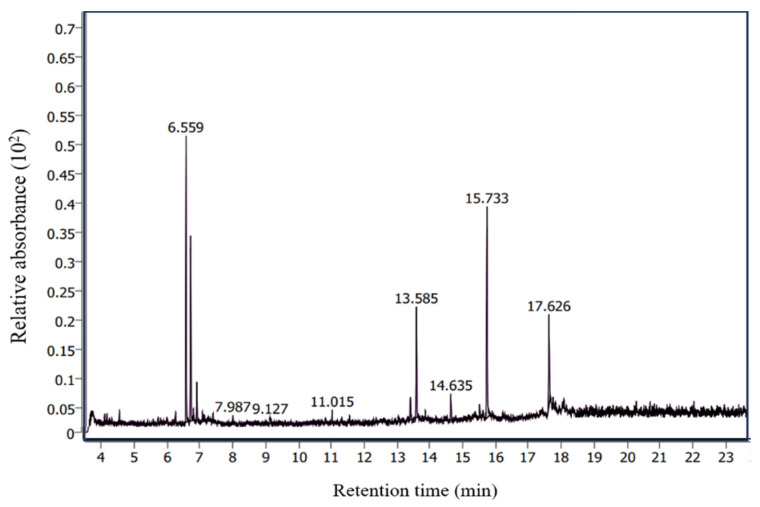
GC-MS chromatogram of the acidic fractions separated from sugarcane leaf ethanol extracts.

**Figure 5 plants-13-02085-f005:**
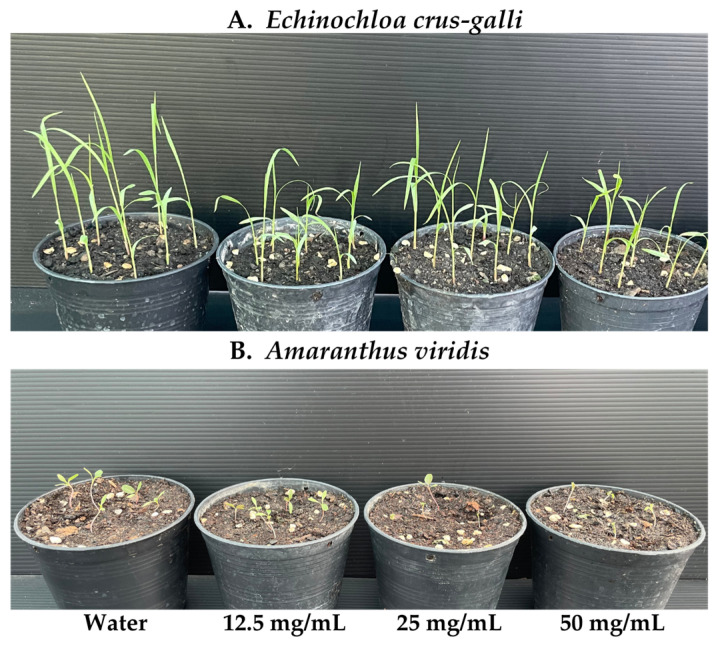
Evaluation of AE formulation herbicidal activity in pot experiments as leaf surface spraying after seed germination. The symptom of *Echinochloa crus-galli* (**A**) and *Amaranthus viridis* (**B**) at 2 days after sprayed application.

**Figure 6 plants-13-02085-f006:**
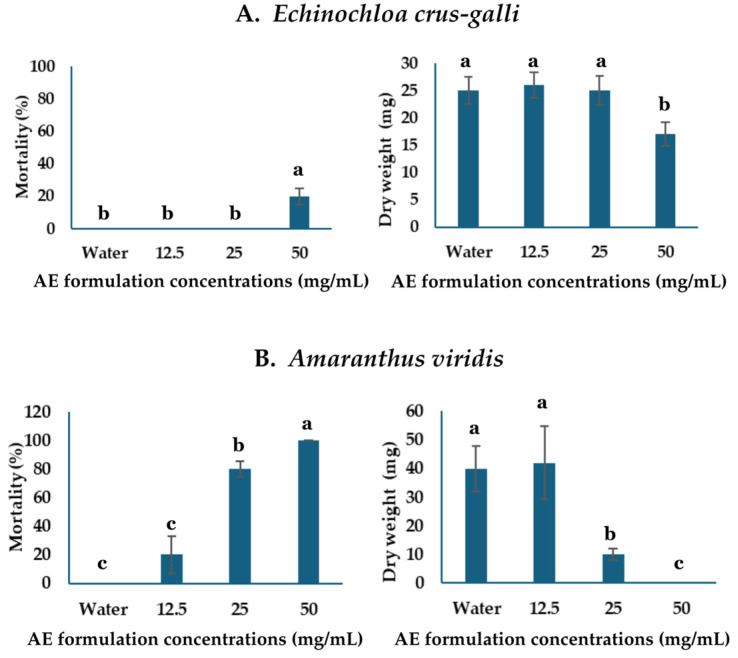
Effect of AE formulation spray application on the mortality and the dry weight of *Echinochloa crus-galli* (**A**) and *Amaranthus viridis* (**B**). The data are the means ± SDs for four biological replicates, and different letters indicate significant differences at *p* < 0.05 (Tukey test).

**Figure 7 plants-13-02085-f007:**
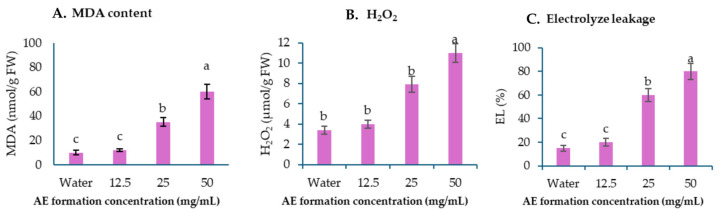
Effects of AE formulation on MDA content (**A**), hydrogen peroxide (**B**), and electrolyte leakage (**C**) in *Amaranthus viridis* leaves at 24 h after spray application.

**Figure 8 plants-13-02085-f008:**
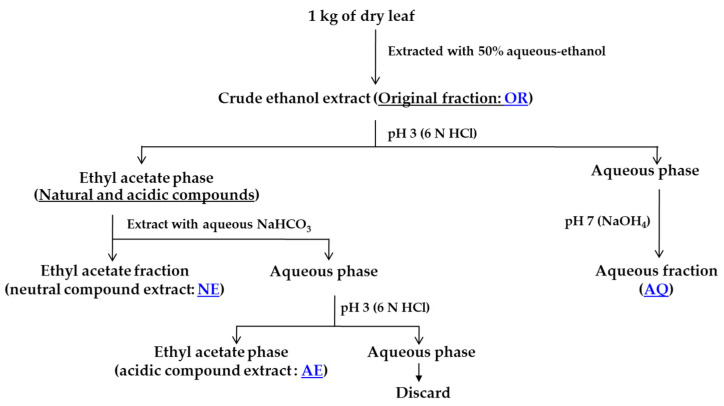
Flow chart for extraction and partial separation of active compounds by acid–base solvent partitioning.

**Table 1 plants-13-02085-t001:** The percentage yield of crude extracts of sugarcane leaves obtained from various solvents and the IC_50_ value of each crude extract on *Amaranthus viridis* seedling growth.

Solvent System (Water:Ethanol [*v*/*v*])	Crude Yield (%)	IC50 Values for *Amaranthus viridis* Seedling Growth (ppm)		
		Seed Germination	Shoot Length	Root Length
100:00	1.35	4997 ^b^	6298 ^a^	5027 ^a^
75:25	4.87	3850 ^a^	6169 ^a^	3864 ^a^
50:50	8.98	4016 ^b^	4819 ^b^	3094 ^a^
25:75	12.92	3434 ^c^	4484 ^b^	2990 ^c^
00:100	10.74	3160 ^c^	3715 ^c^	1565 ^d^

Different superscript letters in a column indicate a significant difference at *p* < 0.05, as determined by Tukey’s HSD test.

**Table 2 plants-13-02085-t002:** Bioactive compounds in the acidic fractions separated from the sugarcane leaf extracts.

	Identified Compound	Formula	Peak Area (%)	Retention Time (min)
**1.**	Iron, tricarbonyl l [N-(phenyl-2-pyridinylmethylene) benzenamine-N, N-]	C_21_H_14_FeN_2_O_3_	24.5	6.559
**2.**	3,5-Dimethyl-1,2,4,3,5-trioxadiborolane,	C_2_H_6_B_2_O_3_	16.19	6.702
**3.**	Propane, 1-[1 [difluoro [(trifluoroethenyl) oxy]methyl]-1,2,2,2-tetrafluoroethoxy]-1,1,2,2,3,3,3-heptafluoro	C_8_F_16_O_2_	1.01	6.787
**4.**	Cyclohexyl methyl S-2-(dimethylamino) ethyl propyl phosphonothiolate	C_14_H_3_NO_2_PS	3.08	6.890
**5.**	Pyridine, 2-chloro-3-fluoro-, 1-oxide	C_5_H_3_ClFNO	0.57	7.106
**6.**	Cyclohexyl methyl silane	C_7_H_16_Si	0.64	7.987
**7.**	3-Methyl pyrazolobis (diethylboryl) hydroxide	C_12_H_26_B_2_N_2_O	0.78	9.127
**8.**	2,6-Bis(thiocyanatomethyl)-4-methylanisole	C_12_H_12_N_2_OS_2_	0.33	9.575
**9.**	Borinic acid, diethyl-, (2-ethyl-1,3,2-dioxaborolan-4-yl) methyl ester	C_9_H_20_B_2_O_3_	0.48	10.812
**10.**	O, O-Dimethyl [1-(4-methyl-1,2,5-oxadiazol-3-ylamino)-1-(2-fluorophenyl) methyl] phosphonate	C_12_H_15_FN_3_O_4_P	0.46	11.308
**11.**	Bis (2-ethylhexyl) hydrogen phosphite	C_16_H_35_O_3_P	2.23	13.401
**12.**	Phenol, 2, 4-bis (1,1-dimethylethyl)-	C_14_H_22_O	10.48	13.585
**13.**	3-(Methylthio) phenyl isothiocyanate	C_8_H_7_NS_2_	2.48	14.635
**14.**	2H-Pyran-2-one, 6-[4,4-bis(methylthio)-1,2,3-butatrienyl]-	C_11_H_10_O_2_S_2_	0.75	15.612
**15.**	Dibenzylamine	C_14_H_15_N	22.96	15.733
**16.**	Benzothiazole, 2-(2-hydroxyethylthio)	C_9_H_9_NOS_2_	10.09	17.626
**17.**	1H-1,2,3,4-Tetrazole, 1-(4-methoxyphenyl)-5-[(phenylmethyl)sulfonyl]-	C_15_H_14_N_4_O_3_S	0.74	18.081
**18.**	Arsenous acid, tris (trimethylsilyl) ester	C_9_H_27_AsO_3_Si_3_	0.62	18.344
**19.**	Caprolactone oxime, (NB)-O [(diethylboryloxy)(ethyl)boryl]-	C_12_H_25_B_2_NO_2_	0.32	20.271
**20.**	Tris (tert-butyldimethylsilyloxy) arsane	C_18_H_45_AsO_3_Si_3_	0.63	21.244

## Data Availability

Data is contained within the article.

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
