# Peer review of "Allelopathic Effects of Sugarcane Leaves: Optimal Extraction Solvent, Partial Separation of Allelopathic Active Fractions, and Herbicidal Activities"

_plants, 2024, doi:10.3390/plants13152085_

Round 1

Reviewer 1 Report (Previous Reviewer 1)

Comments and Suggestions for Authors

Dear editor:

The paper has been improved, but I think it still needs improvement.

Both the introduction and the discussion can be improved.  The introduction should be written more fluently, connecting the concepts more. The discussion should be expanded, adding information on herbicide tests, concentrations, parameters measured .... where it is reflected that the tests carried out are valid.

-In section 3.2 of the results, the concentrations used are 1250 ppm or 1,250 ppm? The text refers to 1,250 and in the figures 1250.

-Figure 3 does not provide anything

-Section 3.3 of the results: the order in which it is mentioned and the table is the opposite of the order in which they are put.

-Figure 5: the references to A. viridis ( C) and E. cruz-galli are confused.

-Conclusions : line 502 y 506 : methanol or ethanol?

Author Response

Comment: Both the introduction and the discussion can be improved.  The introduction should be written more fluently, connecting the concepts more. The discussion should be expanded, adding information on herbicide tests, concentrations, parameters measured .... where it is reflected that the tests carried out are valid.

Response: Thank you so much for your valuable suggestions. We added the introduction in line 111- 115.

Comment: In section 3.2 of the results, the concentrations used are 1250 ppm or 1,250 ppm? The text refers to 1,250 and in the figures 1250.

Response: Thank you so much for your valuable suggestions. We mean 1250 ppm, thus we changed 1,250 ppm to 1250 ppm in the text.

Comment: Figure 3 does not provide anything

Response: Thank you so much for your valuable suggestions. We would like to make visible the various inhibitory effect of different fractions and different concentration effect on bioassay weeds.

Comment: Section 3.3 of the results: the order in which it is mentioned and the table is the opposite of the order in which they are put.

Response: Thank you so much for your comment.  We describe the compounds arranged in a decreasing order based on their mass. We added text of  “arranged in a decreasing order based on their mass” in Line 185

Comment: Figure 5: the references to A. viridis ( C) and E. cruz-galli are confused.

Response: Thank you so much for your comment. We change the reference from A. viridis ( C) and E. crus-galli to A. viridis ( B) and E. crus-galli

Comment: Conclusions : line 502 y 506 : methanol or ethanol?

Response: Thank you so much for your comment. I missed. I mean ethanol. We changed from methanol to ethanol in line 512 and 516

Reviewer 2 Report (Previous Reviewer 2)

Comments and Suggestions for Authors

The material method is not clear. How the extracts were taken. It should be supported by literature. It is difficult to understand how the water/solvent extract is taken. Extracts are taken with solvent and then doses are adjusted. Dose adjustments are complicated. 10 seeds per petri is too few. again, this problem is the same in pot trials. how many ml of water was added to the petri. it is not clear how the seedlings were sprayed. in short, the study planning was not very well done.

Comments on the Quality of English Language

The language should be improved, there are some spelling mistakes, and a native speaker should be able to read and fluentise it

Author Response

Comment: The material method is not clear. How the extracts were taken. It should be supported by literature. It is difficult to understand how the water/solvent extract is taken. Extracts are taken with solvent and then doses are adjusted. Dose adjustments are complicated. 10 seeds per petri is too few. again, this problem is the same in pot trials. how many ml of water was added to the petri. it is not clear how the seedlings were sprayed. in short, the study planning was not very well done.

Response: Thank you so much for your valuable suggestions. The authors would like to explain the answers point by point as provided below:

Comment:  The material method is not clear. How the extracts were taken. It should be supported by literature.

Response: We have described how the extracts were taken in section 4.1 Allelopathic effects of solvent extracts against weeds and the crude extract yields. Also, we inserted the sentence “ Sugarcane leaf powder was extracted to determine their allelopathic effects following the protocol described previously by Thinh et al. [16] ” for more clarity (Line 341-343).

Comment:  It is difficult to understand how the water/solvent extract is taken. Extracts are taken with solvent and then doses are adjusted. Dose adjustments are complicated

Response: The two supernatants were combined, and the solvent was removed in a rotary vacuum evaporator at 40°C to yield the crude aqueous ethanol extracts. The crude sugarcane leaf extracts were dissolved in and diluted with their initial solvents to different concentrations. All necessary information was mentioned in 4.1 Allelopathic effects of solvent extracts against weeds and the crude extract yields.

Comment: 10 seeds per petri is too few. again, this problem is the same in pot trials. how many ml of water was added to the petri.

Response: We used twenty seeds for each test plant per petri. The sentence “ Twenty seeds of A. viridis and/or E. crus-galli were evenly spread onto the paper” was mentioned in Line 353-354. Also, the sentence “ Then, 5 mL of each concentration was added to filter paper (No. 2) in a 90 mm Petri dish” was mentioned in Line 357-358. For pot trials, 10 seedlings at the 2 leaf stages of each test plant were used in the experiments. According to previous literature, they used 10 seedlings and that amount of plant can determine allelopathic effects.  

Comment: it is not clear how the seedlings were sprayed. in short, the study planning was not very well done.

Response: We have described how the seedlings were sprayed in Materials and Methods.  The sentenceTen germinating seeds of either E. crus-galli or A. viridis were planted at a depth of 2 mm. The E. crus-galli and A. viridis seedlings that emerged 7 days after planting, at the 2 leaf stages, were categorized as early post-emergence. The weeds were treated with various concentrations of active ingredients (ai) including 12.5, 25, and 50 mg/mL. The application was performed using a hand nozzle and a water spray at a rate of 750 L/ha. In the control group, distilled water was utilized for applicationwas mentioned in section 4.4 Herbicidal activities of the active fractions (Line 419-424)

Comments on the Quality of English Language:

The language should be improved, there are some spelling mistakes, and a native speaker should be able to read and fluentise it

Response: the manuscript was proved by professional academic by English editing by MDPI No.81154.

Round 2

Reviewer 1 Report (Previous Reviewer 1)

Comments and Suggestions for Authors

I still think there are two aspects of the paper that are deficient. The wording of the introductory paragraph ( line 53-87) is not correct and the discussion lacks quality. These two suggestions have not been considered by the authors. It is up to the academic editor to decide whether it should be modified or published as is.   Sincerely

Author Response

Response to Comments from Reviewer

Comment: I still think there are two aspects of the paper that are deficient. The wording of the introductory paragraph (line 53-87) is not correct and the discussion lacks quality. These two suggestions have not been considered by the authors. It is up to the academic editor to decide whether it should be modified or published as is.  

Response: Thank you so much for your valuable suggestions. We deleted sentences from lines 111 – 115. And we add new sentences in lines 67 - 71 to make a link why we formulate the crude to the suspension concentrate. And we also example other crude extract formulate to the herbicide product in line 74-77.  

In the discussion, we explain which solvent system is suitable for extracting essential substances from a sugarcane leaf. And we explain how to separate primates by this technique. What classes of active substances are found, which vary depending on the type of plant. We also have a biological test, which is a widely accepted practice, and there is an analysis of the important substances on the side of that group. We have processed the raw extract into a mixture that can be applied and explained what the mechanisms of the mixture are due to.

Reviewer 2 Report (Previous Reviewer 2)

Comments and Suggestions for Authors

The requested corrections have not been made

Comments on the Quality of English Language

 Extensive editing of English language required

Author Response

Dear Reviewers/Editor

Comment: The requested corrections have not been made.
Response: I confirm that I answered it in my current response to you.

Comment: Extensive editing of English language required

Response: This article was updated in English by MDPI's service in 90% of the cases, as I have attached the certification.

Round 3

Reviewer 2 Report (Previous Reviewer 2)

Comments and Suggestions for Authors

May be published in final edited form

This manuscript is a resubmission of an earlier submission. The following is a list of the peer review reports and author responses from that submission.

Round 1

Reviewer 1 Report

Comments and Suggestions for Authors

This work does not provide anything new. There are numerous studies showing that the amount of extract is dependent on the solvent used and in different aqueous proportions.  On the other hand, the acid-base solvent partitioning technique yields a fraction which is the most active, but this does not mean that this is necessary to obtain biologically active compounds, and furthermore, it has not been shown that these are the compounds with allelopathic activity.

There are errors such as:

Concentrations are 1200, 2500, 5000 ppm..... or 1,200; 2,500; 5,000 ppm......?

In figure 5, extracted with 50% ethanol-water or with 100% ethanol?

In conclusion they refer to the solvent methanol, is it not ethanol?

In my opinion this paper is not interesting to be published in Plants.

Reviewer 2 Report

Comments and Suggestions for Authors

The article was not studied in a pre-planned manner. a single petri trial was conducted. here the effective dose should have been used in pot trials. At the same time, the basic substances determined as a result of GC-MS analyses should have been tested on weed seeds. especially Iron, tricarbonyl l [N-(phenyl-2-pyridinylmethylene) benzenamine-N, N-], Dibenzylamineibenzylamine, and 3,5-Dimethyl-1,2,4,3,3,5-trioxadiborolane should have been tested for their herbicidal effect on weed seeds. There is a lack of literature. it should be supported by more recent literature.

Comments on the Quality of English Language

Moderate editing of English language required